# Correlation Between Spinopelvic Parameters, Body Mass Index, Waist Circumference, and Chronic Non-Specific Low Back Pain

**DOI:** 10.3390/life15010016

**Published:** 2024-12-27

**Authors:** Daphne Butzen, Yannick Smolders, Tom Stroobants, Gino Verleye, Dieter Thijs, Erik Van de Kelft

**Affiliations:** 1Faculty of Medicine and Health Sciences, University Antwerp, 2610 Antwerp, Belgium; daphne.butzen@student.uantwerpen.be (D.B.); yannick.smolders@student.ua.be (Y.S.); tom.stroobants@studentua.be (T.S.); 2Department of Statistics, University Ghent, 9000 Ghent, Belgium; gino.verleye@ugent.be; 3Department of Statistics, Free University Brussels, 1090 Brussels, Belgium; 4Department of Neurosurgery, Vitaz, 9100 Sint-Niklaas, Belgium; dieter.thijs@vitaz.be; 5Department of Neurosurgery, University Hospital Antwerp, 2610 Antwerp, Belgium

**Keywords:** body mass index, waist circumference, pelvic incidence, low back pain, spinal sagittal balance

## Abstract

Study Design: This is an observational study. Objectives: In general practice, it is noted that some people can deal more easily with a prominent belly than others. Recent use of spinopelvic parameters in the analysis of the spine might explain this observation. This study aimed to determine the correlation between pelvic incidence (PI), waist circumference (WC), and body mass index (BMI) in patients with non-specific chronic low back pain. We hypothesized that people with a low PI (non-pronounced lumbar lordosis) have significantly lower WC values than those with a high PI (pronounced lumbar lordosis). Methods: Adult patients presenting to the outpatient neurosurgery clinic with non-specific chronic low back pain who had undergone full spine radiography were included. The PI, BMI, and WC were measured in all cases. Results: We included 272 patients (male–female ratio, 1.08) with a mean age of 54 years. There was a statistically significant difference (*p* < 0.05) in the mean PI according to BMI group. The mean PI in our population was 57.8° (range 28.4–97.2°, SD 12.1°). A significant correlation coefficient of 0.271 (*p* < 0.001; 95%CI 0.157–0.377) was found between BMI and PI and 0.410 (*p*-value < 0.001; 95%CI 0.262–0.539). Conclusions: We found a significant correlation between PI, BMI, and WC. This finding is the first step in confirming our hypothesis that a patient with a high PI might be able to tolerate being overweight and a high WC better than patients with a low PI, possibly because of their ability to retrovert the pelvis to a greater extent. Further research is warranted to investigate whether people with a high pelvic PI can better cope with obesity, especially those with a higher waist circumference and abdominal weight.

## 1. Introduction

Obesity is a complex and multifactorial disease with multiple unknown correlations, and has become a worldwide healthcare problem [1,2]. All over the world, researchers are trying to evaluate the impact of obesity on health conditions and focus on prevention. The correlation between obesity and chronic low back pain (CLBP), which is also a significant public health burden, has been examined previously [3]. Although the incidence and prevalence of CLBP are high, identification of the responsible pathophysiology and search for a specific pain generator remain difficult [4].

One of the mechanisms contributing to mechanical low back pain is loss of spinal sagittal balance [5]. Characteristic of this problem is that the head no longer projects vertically over the hips. As humans try to maintain a horizontal gaze, sagittal malalignment forces the body to spend more energy maintaining an upright posture. The relationship between balance and energy expenditure was popularized by Dubousset and visualized by his ‘cone of economy’ [6]. The ability to cope with this misalignment depends on the shape of an individual spine. The more lordosis, the easier it is to maintain sagittal balance by retroversion of the pelvis around the hips.

The most widely used spinopelvic parameters are the pelvic incidence (PI), the pelvic tilt (PT), and sacral slope (SS) (Figure 1), as defined by Duval-Beaupère et al. (1998) [7,8]. Another parameter is lumbar lordosis. The higher the PI and SS, the greater the lumbar curvature. Roussouly described four types of lumbar lordosis in an asymptomatic population related to the orientation of the sacrum in the pelvis, which defines the PI [7,9]. Those with a low PI (rather vertical fixed position of sacrum in pelvis) have less possibility of retroverting their pelvis and, thus, to cope with a high waist circumference, than those with a high PI (rather horizontal orientation of the sacrum in the pelvis). Furthermore, the description of these four types of spines includes a strict correlation between PI and the degree of lordosis; the higher the PI, the higher the degree of lordosis (Figure 2).

A high waist circumference is better tolerated by some than by others, who might develop LBP due to being overweight. Spinopelvic parameters may explain this observation [10]. This study aimed to find a correlation between PI, BMI, and waist circumference (WC), to establish an essential step to confirm our assumption. The study population consisted of a cohort of patients with non-specific chronic low back pain (no red flags, no tumor, infection, inflammation, or trauma) analyzed by a full spine X-ray on which the PI was measured. This study did not analyze the pathophysiology of the LBP; we only used the PI values of this cohort in order to find a correlation with BMI and WC. The hypothesis of this study is that people with a higher PI are better able to cope with a high BMI and especially abdominal fat accumulation, perhaps because they can retrovert their pelvis more to stay balanced compared to people with a low PI. As such, we assume that high BMI and WC correlate with high PI, whereas in patients with low PI, high BMI and WC will seldom appear.

## 2. Methods

### 2.1. Study Design

This was a data-gathering cohort study. The sample consisted of 289 patients aged 18–80 years with chronic nonspecific low back pain (CNSLBP). All patients were seen in our out-patient clinic of the Department of Neurosurgery. Patients with a specific CLBP (fracture, infection, tumor, or inflammation) and/or previous spine surgery or pregnancy were excluded from the study. The local ethical committee (Vitaz, St-Niklaas, Belgium) approved the study protocol: 2023-02/EC 23005. All patients have given their written informed consent for the anonymized analysis of PI, BMI, and WC.

### 2.2. Variables and Measurement Methods

The following data were obtained from each patient: age (in years), sex, height (cm), weight (kg), abdominal circumference (cm), and full spine X-ray.

Patients presented with CNSLBP and had already undergone a full-spine X-ray as part of a diagnostic medical imaging workup for their symptoms. The PI was measured on a standing whole-spine radiograph, with the fingertips touching both clavicles (Figure 2a,b). We used the KEOPS (SMAIO, Lyon, France) spinal analysis computer program for automated calculation of spinopelvic parameters.

To identify people with obesity or overweight, the body mass index (BMI) was defined by the National Institute of Health and the World Health Organization (WHO) [11,12,13].

BMI does not consider body fat [8]. To measure waist circumference, we followed the WHO STEPS protocol [11], that instructs the measurement of waist circumference at the approximate midpoint between the lower margin of the last palpable rib and top of the iliac crest. The measurement accuracy depends on three factors. First, the measuring tape is tightened. The tape should be tight around the body, but not pulled so tightly that it is constricting. Second, the patient must stand upright with arms at the sides and feet close together. Finally, the waist circumference should be measured at the end of normal expiration because the amount of space in the lungs depends on the respiration phase, which can affect the waist circumference. We also instructed the patient to relax because a relaxed posture was best for taking the waist circumference [11]. The cut-off for normal waist circumference was 102 cm and 88 cm for men and women, respectively [13]. Patients with missing data were excluded from the study.

### 2.3. Statistics

Data analysis was performed using Statistical Product and Service Solutions (SPSS, version 28). We obtained the mean value, range, and standard deviation (SD) for pelvic incidence (PI), BMI, and waist circumference. The Chi-square test was used to compare categorical variables. To compare the various means, a one-way ANOVA test was performed the various means. To compare the two means, we used Student’s t-test. Pearson’s correlation test was performed to determine the correlation between BMI, waist circumference, and PI. A Spearman correlation test was used to determine the correlation between BMI categories and types of lumbar lordosis (LL). A scatter plot was constructed to visualize a potential nonlinear correlation. Regression analysis was used to define the correlation of the PI with the BMI on the one hand and the waist circumference on the other hand. To measure the influence of PI on BMI and waist circumference, a PATH model was used. A *p*-value < 0.05 was considered as significant.

## 3. Results

During the observation period (01/2020–12/2022), 289 patients with CNSLBP presented at the neurosurgery outpatient clinic in Vitaz Hospital, St-Niklaas, Belgium. After the elimination of double inclusions and patients with missing data, data from 272 patients were analyzed (See patient flow in Figure 3).

The mean age (in years), mean weight (in kg), and weight range (in kg) in the BMI groups were as follows: underweight (54 years, 68 kg, 46–64 kg), normal weight (67 years, 60 kg, 48–90 kg), overweight (61 years, 84 kg, 61–103 kg), and obese (59 years, 99 kg, 68–135 kg). According to sex, 72 men (51%) and 65 women (50%) were categorized as having a low waist circumference, and 69 men (49%) and 66 women (50%) with a high waist circumference. The mean WC (cm) and range were 101 cm (58–136 cm) in men and 90 cm (55–135 cm) in women. The baseline patient characteristics are shown in Table 1.

The mean PI for all patients was 57.8° (range 28.4–97.2°, SD 12.1°). The mean PI according to BMI was 48.7° (95%CI 42.9–54.5°) in the underweight group (10/272 or 4%), 55.0° (95%CI 52.4–57.6°) in the normal-weight group, 57.7° (95%CI 54.9–59.6°) in the overweight group, and 61.7° (59.1–64.3°) in the obese group (Table 2). The same calculation was made for the WC groups: a mean PI of 53.3° in the low WC group and 62.2° in the high WC group.

There was a statistically significant difference in the mean PI between the BMI groups, as determined by one-way ANOVA (F(3, 268) = 6.950, *p* < 0.001). Subgroup analysis showed significant differences between the underweight and normal-weight groups and the obese group (*p* < 0.01). There were no significant differences in the mean PI of the overweight group. A statistically significant difference in the mean PI between the high and low WC groups was found (*p* < 0.001).

A statistically significant correlation was found between BMI and PI of 0.271 (*p* < 0.001; 95%CI, 0.157–0.377) (Table 1). This correlation was further analyzed according to sex. A correlation factor of 0.240 (*p* < 0.005; 95%CI, 0.078–0.390) for males and 0.351 (*p* < 0.001; 95%CI, 0.191–0.493) for females was found. The correlation factor between WC and PI was 0.410 (*p* < 0.001; 95%CI 0.262–0.539) for men and 0.398 (*p* < 0.001; 95%CI 0.244–0.534) for women These results indicate a significant correlation between these factors.

For the relationship between the PI and BMI groups, a significant correlation factor of 0.247 (*p* < 0.001; 95%CI, 0.129–0.359) was found.

## 4. Discussion

In this data-gathering cohort study of patients with chronic non-specific low back pain (CNSLBP), we noted a statistically significant correlation between the mean PI and BMI groups with a correlation factor of 0.247. Subgroup analysis showed significant differences between the underweight and obesity groups, and between the normal-weight and obese groups. No differences in mean PI were observed between the overweight group and the obese and normal-weight groups. We also found a significant correlation between the PI and waist circumference; although this study does not prove that high PI patients can tolerate obesity better, it seems to be a good correlation. Therefore, in a cohort of patients with CNSLBP, high BMI/WC was positively correlated with high PI.

In 2013, the first investigation of a potential correlation between spinopelvic parameters and high waist circumference was reported by Nuttall F.Q. and Romero-Vargas et al. [14,15], but their statistical analysis revealed no significant difference. The study population was smaller and the methods not so clear. There premise, however, that high BMI/WC correlates with high PI, was the same as ours and reflects our experience in daily life, when observing people with high WC; they seem to have a pronounced lumbar lordosis, accentuated by a retroflexion in order to keep a good sagittal balance, despite their belly. Since then, a few other studies have reported a weak correlation between the PI and BMI, but none have reported a statistically significant correlation between the PI and WC. Uysal et al. [16] reported a weak positive correlation between PI and the thickness of abdominal subcutaneous adipose tissue and the thickness of mesenteric adipose tissue. Araújo et al. [17] found that higher BMI and central obesity are important potential determinants of non-neutral posture among adults [17].

All these studies have the same hypothesis, based on daily observation. Most of these studies suggest a positive correlation, but ours, for the first time, confirms this. Based on the results of this study, it can be assumed that patients with a high PI can tolerate a prominent belly more easily than those with a low PI, although this was not proven in this study. Defined by their high PI, these patients have pronounced lumbar lordosis, as their sacrum is oriented horizontally in the pelvis. As such, they have a higher capacity to retrovert their pelvis when they have to cope with an important waist circumference without having problems keeping their spine well-balanced. In contrast, patients with a low PI have their sacrum rather vertically fixed in the pelvis; as such, they have small lumbar lordosis. Therefore, their capacity to retrovert the pelvis to cope with a prominent belly while maintaining a good balance is limited. These results should stimulate further research on how high pelvic incidence can be a risk factor for the development of obesity, as it might give the patient the opportunity to easily cope with it.

In people with a normal BMI but with a high PI, the LL is pronounced; this means that the apex of the LL projects more ventrally and cranially than in people with a low PI. Even without a high WC, people with a high PI, and thus a high LL, present with a more pronounced belly, even with a normal BMI, due to the shape of their spine.

In other studies, the correlation between WC and CNSLBP was measured. Excessive fat mass seemed to be the essence of this correlation [17,18]. These authors did not examine the PI of these patients. In our study, all the patients presented with CNSLBP. Therefore, we did not study the correlations between PI, WC, and CNSLBP. Moreover, this study does not analyze the pain generator nor its intensity. We, on purpose, did not look for a correlation between pain and spinopelvic parameters, nor BMI/WC. This was not the scope of our study. In order to demonstrate what we wanted to evaluate, we needed a large patient cohort with access to their full spine X-rays. In our country, no EC approves full spine X-rays in adult non-symptomatic people, just for research purposes. Therefore, we chose non-specific chronic LBP patients, since, as part of the diagnostic workup, they all undergo full spine X-rays.

Despite the high prevalence of CNSLBP in the elderly, there is limited evidence regarding the factors associated with this symptom. Our study did not help narrow this gap. Based on the correlations found, we suggest that people with a high PI can cope better with a high WC, even when this is painful [19,20].

The strength of our research is the high number of patients included and the prospective data gathering. To the best of our knowledge, this is the first study to demonstrate a significant correlation between PI and WC. This finding is the first essential step in confirming our hypothesis.

The major limitation of our study was that all the included patients reported CNSLBP. Only healthy volunteers would have been exposed to a full-spine X-ray with a significant radiation dose. An ethical committee would probably not have agreed to include healthy volunteers in this study. It is possible that people who can retrovert the pelvis more to maintain sagittal balance do not have as much LBP, even with overweight. To prove this, however, we should consider a random cohort of the general population, select obese patients who have lumbar pain, and then analyze if PI has any association. As stated before, in our country, no EC will approve exposure to full spine X-ray in an asymptomatic population.

The dispersion measures from our population are not compared with an asymptomatic study population, for example, in Roussouly’s publication ^9^, which makes it impossible to look at the similarities and estimate the whole population. This was not the scope of our study but could give rise to further research.

When similar research is carried out in the future, it would be interesting to measure the respective pelvic tilt (PT) in order to analyze the capacity of pelvic retroversion in cases of obesity.

## 5. Conclusions

Based on our study results, we only found patients with a high BMI and especially a high WC in the higher PI groups, suggesting that, even when painful, these patients can cope more easily with being overweight, probably because of their ability to retrovert their pelvis more than patients with a low PI.

## Figures and Tables

**Figure 1 life-15-00016-f001:**
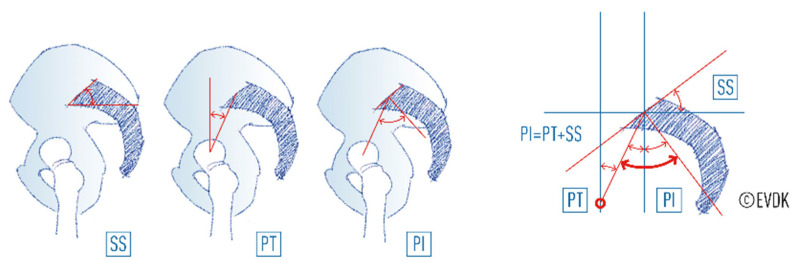
Definition of spinopelvic parameters PI, PT, and SS.

**Figure 2 life-15-00016-f002:**
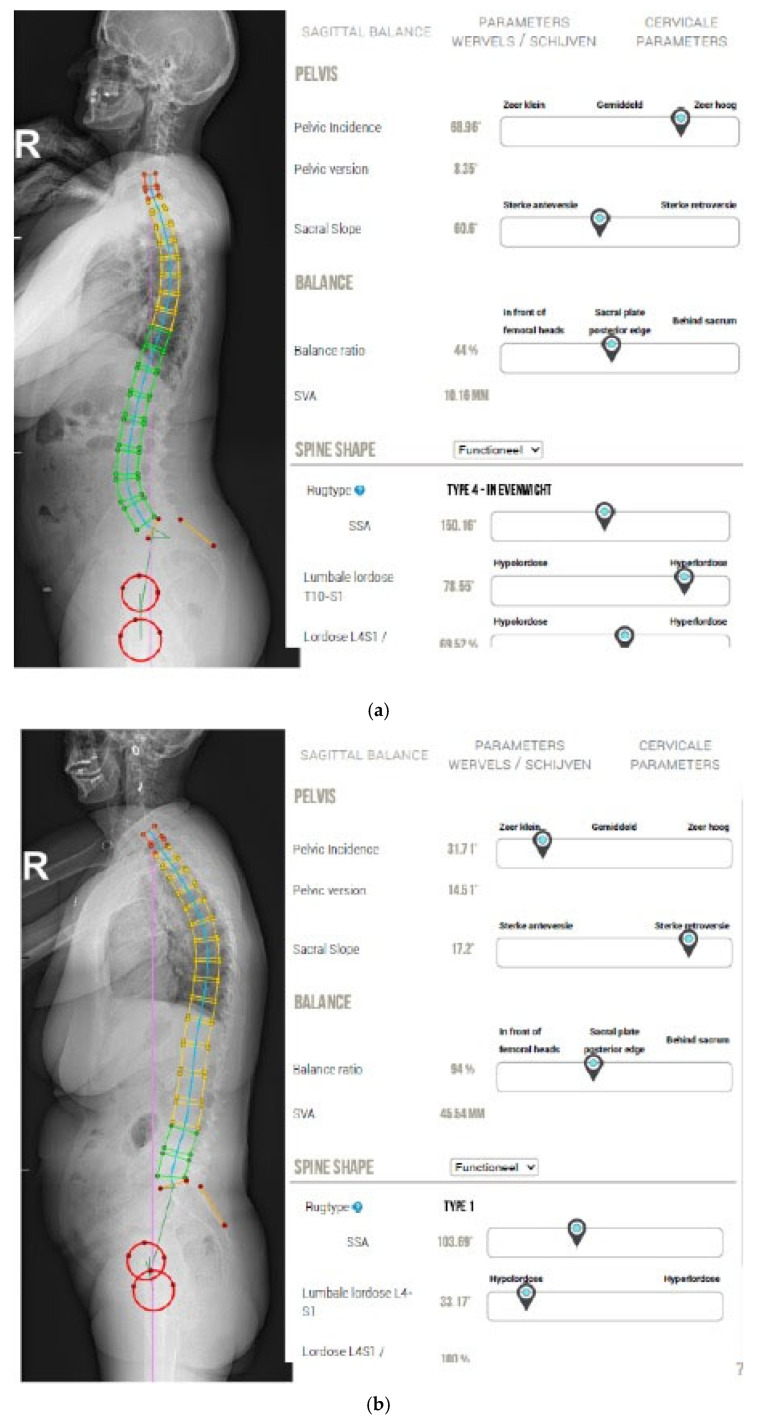
(**a**) X-ray of patient with high PI; (**b**) X-ray of patient with low PI.

**Figure 3 life-15-00016-f003:**
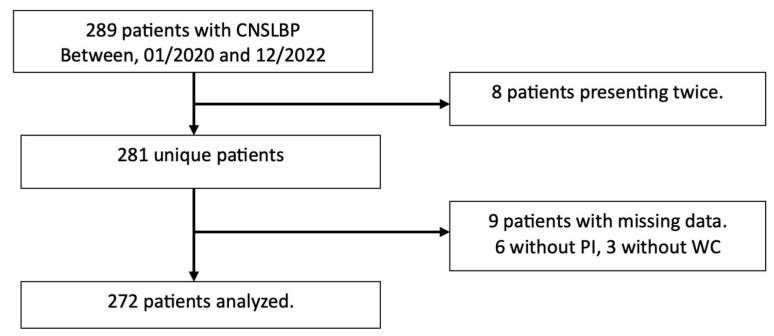
Patient flow.

**Table 1 life-15-00016-t001:** The mean PI according to the BMI group.

Variable	Under-Weight (SD)	Normal Weight, Mean (SD)	Over Weight, Mean (SD)	Obesity, Mean (SD)	Spearman Correlation Factor	*p*-Value
PI	48.7	55.0	57.7	61.7	0.247	<0.001

**Table 2 life-15-00016-t002:** Demographics (high WC: WC < 102 cm (m) or <88 cm (f); low WC: WC ≥ 102 cm (m) or ≥88 cm (f)).

Variable		Number of Participants	% of Participants	Mean	Range	Standard Deviation
Sex	Male Female	141 131	51.8 48.2			
Age				60	22–93	13
BMI	Underweight Normal weight Overweight Obesity	10 78 93 91	3.7 28.7 34.2 33.5			
WC	High Low	137 135	50.4 49.6			
PI				57.8	28.4–97.2	12.1

## Data Availability

Data are available upon request to the corresponding author.

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
