# Peer review of "Correlation Between Spinopelvic Parameters, Body Mass Index, Waist Circumference, and Chronic Non-Specific Low Back Pain"

_life, 2024, doi:10.3390/life15010016_

Round 1

Reviewer 1 Report

Comments and Suggestions for Authors

In the line 74 it is mentioned that: ʺ The sample consisted of 289 patients aged 18 years with chronic nonspecific low back pain (CNSLBP)ʺ. Do you consider that all patients are 18 years old?

 Please add a table with parameters according to categories of sociodemographic and occupational characteristics (Blue collar vs. white collar).

Author Response

Comments 1; In the line 74 it is mentioned that: ʺ The sample consisted of 289 patients aged 18 years with chronic nonspecific low back pain (CNSLBP)ʺ. Do you consider that all patients are 18 years old?

We are very grateful for this comment as we forgot to indicate the upper age limit for inclusion; 80 yrs. We changed this in the text at line 74 in the 2. Methods section.

Comments 2;  Please add a table with parameters according to categories of sociodemographic and occupational characteristics (Blue collar vs. white collar).

We would like to thank the reviewer for this comment. Unfortunately, we have not this information systhematically in our electronic patient files. Furthermore, this study is not carried out to observe LBP, but rather to analyze the combination of PI and BMI. As such, we think, for the purpose of this study, the sociodemographic nor occupational characteristics are of any importance.

Reviewer 2 Report

Comments and Suggestions for Authors

The authors present a study on: Correlation between spinopelvic parameters, body mass index, waist circumference 2 and chronic non-specific low back pain. The topic is interesting, however, a thorough analysis must be done.

First of all, the abstract must be modified, there is no introduction and all the information is disordered. Keywords ok.

In the introduction he starts talking about obesity and goes on to say that tobacco is the main cause of death... I don't make sense of it... The introduction should be modified. It should include the study population, risk factors and possible correlations in a structured manner.

The methods should indicate where the sample was recruited from, if the patients have given their consent.

The results are correctly presented.

The discussion is too brief, it does not provide many correlations with previous studies, please expand this section.

Author Response

Comments 1 :First of all, the abstract must be modified, there is no introduction and all the information is disordered.

Response 1 : We would like to thank the reviewer for comments and suggestions. we did our best to take them into account to make this paper better.

In line with your comment, we modified the abstract with an introduction and ordering of the information. We changed it in the text.

Comments 2: In the introduction he starts talking about obesity and goes on to say that tobacco is the main cause of death... I don't make sense of it... The introduction should be modified. It should include the study population, risk factors and possible correlations in a structured manner.

Response 2: We changed the intro according to your suggestions. The reference to smoking has been deleted and we described the study population in a structured manner. we hope these changes make the intro more clear than before.

Comments 3 : The methods should indicate where the sample was recruited from, if the patients have given their consent.

Response 3 : The study popyulation was described in more detail, including the recuitement. We added the written informed consent of the patients and for what the consent was given.

Comments 4 : The results are correctly presented.

Comments 5 : The discussion is too brief, it does not provide many correlations with previous studies, please expand this section.

Response 5 : We expanded the discussion according to your suggestions. we thank you for this.

Reviewer 3 Report

Comments and Suggestions for Authors

For authors

It is an interesting topic.

It's a study that I don't think is finished. The data doesn't seem to be laid out properly and the font used isn't the same. There are a few things:

Lines 28-29: Keywords: Body Mass Index; Waist circumference; Pelvic Incidence; weight; low back pain; radiograph; Lumbar Lordosis; spinal sagittal balance; obesity; spine

I think there are too many keywords, there should be a maximum of 5

I don't think Table 1 with the BMI classification needed to be included. It already presents very well-known things.

Lines 123-124: Our study included 10 underweight, 78 normal-weight, 93 overweight, and 91 obese patients

The data from the text can be found in Table 3. I don't think this needs to be repeated. The table is easy to read.

Lines 151-153: These results indicate a significant correlation between these factors. For the relationship between the PI and BMI groups, a significant correlation factor of 0.247 (p < 0.001; 95%CI, 0.129–0.359) was found.

There is a difference in the font.

We also find it in lines 162-163:Therefore, in a cohort of patients with CNSLBP, high BMI/WC was positively correlated with high PI.

Same on lines 172-174 :Based on the results of this study, it can be assumed that patients with a high PI can tolerate a prominent belly more easily than those with a low PI, although this was not proven in this study

And on line 184:and thus a high LL, present with a more pronounced belly, 184 even with a normal BMI.

Same on lines 188-191:In our study, all the patients presented with CNSLBP. Therefore, we did not study the correlations between PI, WC, and CNSLBP. Despite the high prevalence of CNSLBP in the elderly, there is limited evidence regarding the factors associated with this symptom. Our study did not help narrow this gap

There are many more such examples.

Line 170 Araújo et al. found that higher B

What is the reference number? In the text, if you put a name, put the reference number next to it.

Lines 165-166 Romero-Vargas et al., but their statistical  analysis revealed no significant difference. 16

But Romero-Vargas is at number 17 in references

Line 168 Uysal et al. 17

But Uysal is at number 18 in references

There are more examples

It's annoying when reading that the references are not listed in parentheses. And if a name is listed, it doesn't match the number in the references.

There are presented shortcomings determined by the fact that other correlations were not studied and that there was no comparison group

I think there is still work to be done on this study, it seems unfinished.

Further studies are certainly needed.

My comments are only intended to make the paper better. Good luck!

Author Response

First of all, we would like to thank the reviewer for his comments and suggestions to make this paper better. In general, we chacked for proper use of the same font, and checked for correct referencing and changed it in the text.

Comments 1: Lines 28-29: Keywords: Body Mass Index; Waist circumference; Pelvic Incidence; weight; low back pain; radiograph; Lumbar Lordosis; spinal sagittal balance; obesity; spine

Response 1 : we reduced the number of Keywords to 5, as you suggested.

Comments 2 : I don't think Table 1 with the BMI classification needed to be included. It already presents very well-known things.

Response 2 : We deleted this Table

Comments 3 : Lines 123-124: Our study included 10 underweight, 78 normal-weight, 93 overweight, and 91 obese patients

Response 3 : We deleted the by you indicated lines.

Comments 4 : There are presented shortcomings determined by the fact that other correlations were not studied and that there was no comparison group

Response 4 : When studying low back pain, things are very quickly getting complicated. Therefore, we choose jyst three eaysy to measure parameters, very objective, to investigate our H0; PI, BMI and WC. the results of our study confirm our experience in dauly life; people with a pronounced belly seem to have an important lumbar lordosis and have the capacity to retroverd, therby restoring their sagittal balance.

Of course, much more correlations can be subject of further study, and we are kean to continue these observations. however, for the clarity, we intentionally wanted to keep this basic study as simple as possible, but with a large sample size, in order to make a statement, statistically relevant.

Round 2

Reviewer 2 Report

Comments and Suggestions for Authors

The authors have attended to all the recommendations proposed above. I don't have any comments for additional improvements.